# Sphingomyelins Prevent Propagation of Lipid Peroxidation—LC-MS/MS Evaluation of Inhibition Mechanisms

**DOI:** 10.3390/molecules25081925

**Published:** 2020-04-21

**Authors:** Giulia Coliva, Mike Lange, Simone Colombo, Jean-Pierre Chervet, M. Rosario Domingues, Maria Fedorova

**Affiliations:** 1Institute of Bioanalytical Chemistry, Faculty of Chemistry and Mineralogy, University of Leipzig, Deutscher Platz 5, 04103 Leipzig, Germany; giulia.coliva@uni-leipzig.de (G.C.); mike.lange@uni-leipzig.de (M.L.); 2Center for Biotechnology and Biomedicine, University of Leipzig, Deutscher Platz 5, 04103 Leipzig, Germany; 3Mass Spectrometry Center, LAQV-REQUIMTE, Department of Chemistry, University of Aveiro, Campus Universitário de Santiago, 3810-193 Aveiro, Portugal; simone.c712@gmail.com (S.C.); mrd@ua.pt (M.R.D.); 4CESAM, ECOMARE, Department of Chemistry, University of Aveiro, Campus Universitário de Santiago, 3810-193 Aveiro, Portugal; 5Antec Scientific, 2382 NV Zoeterwoude, The Netherlands; jp.chervet@antecscientific.com

**Keywords:** sphingomyelins, lipid peroxidation, liposomes, electrochemical oxidation, oxidative stress, LC-MS

## Abstract

Free radical driven lipid peroxidation is a chain reaction which can lead to oxidative degradation of biological membranes. Propagation vs. termination rates of peroxidation in biological membranes are determined by a variety of factors including fatty acyl chain composition, presence of antioxidants, as well as biophysical properties of mono- or bilayers. Sphingomyelins (SMs), a class of sphingophospholipids, were previously described to inhibit lipid oxidation most probably via the formation of H-bond network within membranes. To address the “antioxidant” potential of SMs, we performed LC-MS/MS analysis of model SM/glycerophosphatidylcholine (PC) liposomes with different SM fraction after induction of radical driven lipid peroxidation. Increasing SM fraction led to a strong suppression of lipid peroxidation. Electrochemical oxidation of non-liposomal SMs eliminated the observed effect, indicating the importance of membrane structure for inhibition of peroxidation propagation. High resolution MS analysis of lipid peroxidation products (LPPs) observed in in vitro oxidized SM/PC liposomes allowed to identify and relatively quantify SM- and PC-derived LPPs. Moreover, mapping quantified LPPs to the known pathways of lipid peroxidation allowed to demonstrate significant decrease in mono-hydroxy(epoxy) LPPs relative to mono-keto derivatives in SM-rich liposomes. The results presented here illustrate an important property of SMs in biological membranes, acting as “biophysical antioxidant”. Furthermore, a ratio between mono-keto/mono-hydroxy(epoxy) oxidized species can be used as a marker of lipid peroxidation propagation in the presence of different antioxidants.

## 1. Introduction

Lipid peroxidation is a degradation process commonly initiated by reactive oxygen species (ROS) [1] and correlated to several pathologies, such as atherosclerosis [2], neurodegenerative diseases [3], type 2 diabetes [4], chronic inflammation [5], and age-associated diseases [6]. Polyunsaturated fatty acids (PUFAs) are the preferred targets of lipid peroxidation since hydrogens in the bis-allylic position of pentadienyl moieties are highly susceptible for radical abstraction [7]. Over the last years, the oxidation products have been extensively studied for free PUFAs and several phospholipid classes [8], as potential biomarkers of oxidative stress.

Sphingomyelins (SMs) are important constituents of the human lenses [9], the myelin sheath, and of the outer leaflet of the mammalian cell membrane [10]. SMs consist of a sphingoid base linked via amide bond to fatty acyl chain, and a phosphocholine head group linked to a sphingoid base. The amide and the hydroxyl groups of a sphingoid base enables H-bond formation, providing SMs with unique physical characteristics with respect to other membrane phospholipids [11,12,13]. Multiple studies demonstrated the formation of intermolecular H-bonds between SM molecules, involving the amide group and the carbonyl, or the hydroxyl and the carbonyl, conferring lower lateral diffusion, major order, and rigidity in SM-rich membranes [14,15,16,17].

SMs distribution in plasma membrane is not homogeneous. SMs, in association with cholesterol, are known to form membrane microdomains commonly called lipid rafts [18,19]. High SM content, and thus ability to form dense H-bond network between SMs and SM and cholesterol, lead to phase segregation with the formation of a liquid ordered phase [Lo] in lipid rafts, in contrast to a liquid disordered phase [Ld] in the main membrane domain. [20,21,22]. The exact nature of lipid raft formation and their role in the membrane is still controversial and not fully understood [23]. It was hypothesized that the SM content in the membrane might influence lipid susceptibility to oxidative stress [24,25], therefore the inhomogeneous SM distribution could lead to a different predisposition to lipid peroxidation of the various membrane areas. Using liposomes of different lipid compositions, it was demonstrated that higher SM content correlates with lower amount of formed conjugated dienes, markers of oxidized PUFA chains [26]. Furthermore, lower levels of lipid oxidation were detected in cells with a higher SM concentration [27].

Despite high SMs abundance in some tissues and intracellular compartments, only few studies addressed their oxidation products [28,29]. The lack of the data on SM oxidation is partially explainable by the fact that the main SMs detected in nature are saturated, and consequently not a primary target of oxidation. However, the possibility of formation of biologically relevant SM-derived lipid peroxidation products (LPPs) cannot be excluded, since, even if in lower abundance, SMs composed by unsaturated fatty acids (FA) are known to be present in biological systems. For instance, FA(16:1), FA(18:1), FA(18:2), and FA(24:1) are among the most common unsaturated FAs known to be part of mammalian SMs.

Here, we studied the mechanistic aspects of the SM protective role against lipid peroxidation in liposomes containing different ratios of SM and 1-palmitoyl-2-linoleoyl-*sn*-glycero-3-phosphocholine (PLPC). Using reverse phase chromatography coupled online to high-resolution mass spectrometry (MS), we characterized SM- and PLPC-derived LPPs and followed up their formation up to 96 h post oxidation induction. A detailed molecular characterization of formed LPPs allowed us to demonstrate the redirection of lipid peroxidation pathway from chain propagation, yielding lipid hydroxides/epoxides, towards chain termination, with the formation of corresponding keto derivatives, in the presence of increasing amounts of SMs.

## 2. Results and Discussion

### 2.1. Total Oxidation of SM/PC Liposomes

To monitor the effect of SM content on lipid peroxidation, liposomes with different ratios of SM and PC lipids were created, characterized, and used for in vitro oxidation. Formation of tightly ordered membranes with decreased surface area due to the H-bond formation was previously reported for SM-rich liposomes [30,31]. As expected, liposomes with highest SM fraction had the smallest radius (56 nm for 100 mol% SM liposomes) which gradually increased with lowering SM content (109 nm for 100 mol% PC liposomes; Figure 1A).

Lipid oxidation was monitored by LC-MS/MS every 24 h for a total of 96 h, and peak areas of oxidized species identified by tandem mass spectrometry were compared with the peak areas of the parent lipids (Figure 1B,C). Total oxidation was expressed as a sum of the main LPPs for each lipid class (for detail see Appendix A). The results illustrated the dependence of LPP formation rates on a molar fraction of SM in model liposomes. Thus, total SM oxidation reached 35.5% relative to unoxidized SM in liposomes with 25 mol% of SM, whereas it was only 15.4% in liposomes with 75 mol% of SM. No LPPs were detected in liposomes consisting only from SM (Figure 1B). PC-derived LPPs were more abundant in all studied conditions (Figure 1C). Highest oxidation yield was observed in liposomes consisting only from PLPC (up to 1471.5% at 72 h) and gradually decreased with increasing SM content. Thus, in the presence of 75 mol% of SM, the total amount of PC-derived LPPs corresponded to 170.6% relative to unmodified PLPC (Figure 1C). These results illustrate the inhibitory effect of SM on the initiation and progression of lipid peroxidation in model liposomes with different ratios of SM and PC lipids.

### 2.2. Analysis of LPP Molecular Species Formed in SM/PC Liposomes

For a more detailed view on the process of lipid peroxidation and LPP formation in liposomes of different compositions, the relative abundance of analyzed LPPs was plotted over the exposed time for each liposome composition. Moreover, optimized RP chromatography allowed to separate isomeric species, such as hydroperoxides and dihydroxy-derivatives of PLPC.

Figure 2 illustrates the progression of lipid peroxidation over 96 h in liposomes containing 75 mol% of SM exemplified for PLPC+2O species. Extracted ion chromatograms for *m/z* 790.5598 (±5 ppm) and corresponding tandem mass spectra (Appendix A) allowed the identification of two isomeric species corresponding to PLPC+OOH (RT range 12–13 min) and PLPC+2OH (RT range 10–11 min) derivatives of PLPC oxidized on linoleic acid. At the time point 0, low amounts of PLPC+OOH can be already observed, reaching a maximum after 24 h of oxidation. Although low intensity PLPC+2OH can be detected at 24 h, the proportion of dihydroxy derivatives became noticeable at 48 h, where their signal intensities were comparable to hydroperoxy-derivatives. After 72 and especially 96 h, dihydroxy-PLPC became the most abundant LPP for these isomeric species.

Formation of SM- and PC-derived LPPs was monitored using similar analytical workflow for other oxidized species including lyso-lipids, hydroperoxides, hydroxy(epoxy) and keto derivatives, as well as truncated forms generated via oxidative cleavage of unsaturated fatty acyl chains (Figure 3 and Figure 4).

Among SM-derived LPPs, mono-oxygenated species (hydroxy(epoxy) and keto derivatives) were the most abundant, reaching up to 15.5% and 13.3% in liposomes containing 25 mol% SM, respectively (Figure 3). As it was shown above, LPP abundance as well as the formation rate showed clear dependence on the molar ratio between SM and PC lipids. Thus, abundance of hydroxy(epoxy)- and keto-SM in liposome with 75 mol% of SM corresponded only to 6.1% and 6.9%, respectively. A similar trend was observed for dihydroxy-, diketo-, and keto-hydroxy(epoxy)-SM LPPs, although present at much lower amounts. SM oxidation products formed by truncation at C9 of oleic acid (corresponding aldehyde and carboxylic acid) as well as lyso-derivative formed by the loss of fatty acyl chain were identified and quantified as well. However, their impact, although reproducing the trend for negative correlation of LPP abundance with SM content, was negligible (below 1% relative to unmodified SM).

PC-derived LPPs were more abundant relative to their parent lipid (Figure 4). Here, the most abundant species were represented by LPPs formed by linoleic acid truncation at C9 with the formation of the corresponding aldehyde and carboxylic acid, as well as long chain LPPs formed by the addition of two oxygen atoms (dihydroxy- and keto-hydroxy(epoxy)-PLPC). A clear impact of SM liposomal content on the dynamic of PLPC peroxidation can be observed here as well for all detected oxidized lipids including lysoPC, truncated forms, and mono- and di-oxygenated species. Moreover, not only LPP abundance but also the rates of their formation were decreased by increasing the molar fraction of SM lipids in model liposomes. Thus, a clear increase in the lag phase of LPP formation can be observed almost for all plotted curves when compared between different liposome preparations used for the oxidation.

The results presented above demonstrate that the kinetic of formation and abundance of main LPP types including oxygen addition (hydroxy(epoxy)-, keto-, hydroperoxyl-, dihydroxy-, and keto-hydroxy-derivatives) and oxidative cleavage (aldehydes and corresponding carboxylic acids) products were dependent on liposome composition, and showed negative correlation with the content of SM lipids. That was also true for both SM- and PC-derived LPPs, although PLPC showed much higher oxidation. This effect can be attributed to the formation of the dense network of H-bonds in liposomes with high SM ratio, limiting lipid peroxidation chain reaction.

### 2.3. Electrochemical Oxidation of SM Liposomes

To verify that inhibitory effect of SM on propagation of lipid peroxidation is attributed to its H-bonding properties in liposomes, we performed oxidation of non-liposomal SM (methanolic solution), using electrochemical (EC) cell coupled on-line to MS. EC oxidation was performed using boron doped diamond electrode, for which the OH radical was reported as the main oxidizing species, thus mimicking the condition of in vitro oxidation by the Cu^2+^/ascorbate system used for liposome oxidation. Using a methanolic solution of SM which prevents liposome formation, a much higher abundance of SM oxidation products was obtained in comparison with liposomal SM (Appendix A). Among the most abundant SM-derived LPPs were hydroxy(epoxy)-(72.2% relative to unmodified SM), keto-hydroxy- (33.3%), and lyso-SM (24.2%) derivatives. Even highly oxygenated species represented by addition of three and two oxygen atoms were formed with the relative abundance of up to 10% using EC oxidation of non-liposomal SM, illustrating the absence of SM protective properties against lipid peroxidation in non-liposomal solutions. This illustrates the crucial role of the SM H-bond network in membranes for the inhibition of lipid oxidation propagation.

### 2.4. Mechanistic Aspects of SM Protection against Lipid Peroxidation Chain Reaction

We demonstrated that in the absence of the H-bond network, SM(d18:1/18:1), despite absence of highly oxidizable moieties (e.g., pentadienly moieties in PUFA) undergoes rapid oxidation, which is drastically inhibited in liposomal SMs. Consistent with previously published results on SM-containing liposomes and lipoproteins where lipid oxidation was monitored via specific absorbance of conjugated dienes [26], the dynamic of LPP formation in liposomes with a different SM content demonstrated both propagation of the lag phase and decrease in the slope for all monitored LPPs. Some studies compared the inhibitory effect of SM-rich liposomes on lipoxidation of unsaturated PC lipids with liposomes containing dipalmitoyl PC (DPPC). Interestingly, using absorbance of conjugated dienes as the read out of lipid oxidation, both SM and DPPC showed similar inhibition of PLPC oxidation [26]. However, when an MS-based method was used to monitor lipid oxidation of stearyl-arachidonoyl-PC (SAPC) liposomes, the presence of SM resulted in much lower oxidation rates than DPPC, indicating the importance of the SM-specific H-bond network in preventing radical propagation [25].

Here, using high-resolution MS allowing us to monitor the whole variety of LPPs independent on the presence of conjugated dienes in their structure, we could demonstrate an apparent shift in the pattern within formed LPPs. Thus, at lower SM content, both SM and PLPC mono-oxygenated species were dominated by hydroxy(epoxy)-LPPs (lipid + O). Increased SM content led to a decrease in hydroxy(epoxy)-derivatives relative to keto- (lipid + O - 2H), providing higher keto/hydroxy(epoxy)-LPP ratio for liposomes with a higher SM fraction (Figure 5A).

Identified LPPs were mapped to the known pathways of lipid peroxidation described for mono- and di-unsaturated fatty acids in the presence of transition metals (Figure 5B) [32,33,34,35,36]. Known pathways of keto-derivative (lipid + O - 2H) formation includes the Russel reaction between two peroxyl radicals (*reaction 14*), reaction of lipid peroxyl radical with lipid peroxide (*reaction 15*), and oxidation of alkoxyl radical by metals at higher transition state (e.g., Cu^2+^; *reaction 7*). On the other hand, hydroxy and isomeric epoxy derivatives (lipid + O) are formed from peroxyl radicals via the Russel reaction (*reaction 14*), peroxyl radical addition to unmodified lipid double bond followed by homolytic substitution yielding epoxy LPPs and alkoxyl radical (*reaction 5*), or by alkoxyl radical H-abstraction from adjacent unmodified lipid (*reaction 6*). Except the Russel reaction yielding equimolar amounts of keto and hydroxy derivatives, two other reactions of mono-hydroxy(epoxy) LPP formation are crucial steps in the propagation of lipid peroxidation. Thus, the higher hydroxy(epoxy)-LPP production relative to keto-derivatives observed here for conditions with low SM fraction is indicative for high propagation rates of lipid peroxidation. The decrease of hydroxy(epoxy)-LPP formation observed in SM-rich liposomes would mark an overall decrease in propagation rates. This effect can be explained both by the formation of the SM H-bond network preventing lateral radical propagation by decreasing radical penetration into liposome and their contact sites with metal ions, as well as the lower accessibility of pentadienyl moieties in adjacent lipids in SM(d18:1/18:1)-rich liposomes (propagation constant k_p_ for oleic acid is an order of magnitude lower than for linoleic acid). Thus, the ratio between mono-oxygenated keto and hydroxy/epoxy derivatives can serve as a marker of free radical lipid peroxidation propagation rates.

## 3. Material and Methods

### 3.1. Materials

1-palmitoyl-2-linoleoyl-*sn*-glycero-3-phosphocholine (PLPC) and *n*-oleoyl-d-erythro-sphingosylphosphorylcholine (SM(d18:1/18:1)) were obtained from Avanti Polar Lipids (AL, USA). Chloroform was purchased from Merck (Darmstadt, Germany). Water was purified (resistance > 18 mΩ/cm) on a PureLab Ultra Analytic System (ELGA Lab Water, Celle, Germany). Ammonium bicarbonate, ammonium formate, sodium ascorbate, and copper (II) sulfate anhydrous were purchased from Sigma-Aldrich (Munich, Germany). UPLC-grade methanol, acetonitrile, formic acid, and isopropanol were obtained from Biosolve VB (Valkenswaard, The Netherlands).

### 3.2. Electrochemical Oxidation

Electrochemical oxidation was performed using a ROXY Potentiostat (Antec Scientific, Zoeterwoude, The Netherlands) equipped with a µPrepCell 2.0 containing a Magic Diamond™ working electrode (boron doped diamond). The oxidation was achieved by applying a ramping voltage (2.8 to 3.2 V—in steps of 20 mV/s) at 37 °C and the products obtained were analyzed by on-line coupling of the electrochemical cell with an ESI-HCT Ion Trap MS (Bruker, Bremen, Germany); 25 µmol/L of SM(d18:1/18:1) standard in MeOH:20 mM ammonium formate (1:1 *v/v*) was infused at the flow rate of 5 µL/min to the ROXY EC cell directly coupled to the ESI-MS. The electrospray voltage was set at 4.2 kV; the capillary temperature at 300 °C; sheath gas pressure was 20.00 psi; drying gas flow rate was 5.00 standard liter per minute. The analysis was done in positive mode, and the most intense ions were selected for MS/MS analysis from each full-scan mass spectrum, followed by the dynamic exclusion for 20 min. MS/MS spectra were acquired in profile mode; isolation width was 4.00 *m/z* units, and fragmentation time was 40 ms. Data were analyzed using Compass Data Analysis (version 4.2) and Compass Data Analysis Viewer (version 4.2) (Bruker, Billerica, MA, USA).

### 3.3. Liposome Model and Radical-Induced Oxidation

1-palmitoyl-2-linoleoyl-*sn*-glycero-3-phosphocholine (PLPC) and SM(d18:1/18:1) were mixed in different ratios (0 mol% of SM + 100 mol% of PLPC, 25 mol% of SM + 75 mol% of PLPC, 50 mol% of SM + 50 mol% of PLPC, 75 mol% of SM + 25 mol% of PLPC, and 100 mol% of SM + 0 mol% of PLPC) and dried. For liposome preparations, lipid chloroform solutions (10 mg/mL) were mixed as follows: for 0 mol% of SM + 100 mol% of PLPC liposomes—34.1 µL of PLPC, for 25 mol% of SM + 75 mol% of PLPC liposomes—8.2 µL of SM + 25.6 µL of PLPC, for 50 mol% of SM + 50 mol% of PLPC liposomes—16.4 µL of SM + 17.0 µL of PLPC, for 75 mol% of SM + 25 mol% of PLPC liposomes—24.6 µL of SM + 8.5 µL of PLPC, and for 100 mol% of SM + 0 mol% of PLPC liposomes—32.8 µL of SM.

Dried lipid mixtures were resuspended in 240 mL of 3 mM ammonium bicarbonate (ABC) by vortexing and tip-sonication on ice for 1 min at 30% amplitude using a Vibra-Cell™ tip sonicator (Sonics & Materials, Inc., Newtown, CT, USA) to create the vesicles which were used for oxidation; 30 μL of CuSO4 (750 µmol/L) and 30 μL of ascorbic acid (1.5 mmol/L) were added to the solution to reach the final concentrations of 1.5 mmol/L of total lipid content, 75 µmol/L of CuSO4, and 150 µmol/L of ascorbic acid. The oxidation proceeded for 96 h.

The liposomes’ diameter was determined by a dynamic light scattering using DynaPro NanoStar™ (Wyatt Technology, Dernbach, Germany) at time 0 prior to oxidation. Lipid mixtures were created as described before. To obtain the same final total lipid concentration (1.5 mmol/L), dried lipids were resuspended in 300 μL of 3 mM ABC. Liposomes were centrifuged (20,000× *g*, 10 min) and 10 µL of the supernatant were measured at 663.61 nm (37 °C, 50 acquisitions).

To monitor the oxidation process, aliquots were collected and analyzed every 24 h by LC-MS/MS. Lipids were diluted in isopropanol to achieve a concentration of 4 ng of total lipid in 1 µL of IPA for MS analysis.

### 3.4. LC-MS/MS Analysis

Chromatographic separation was performed using a Vanquish Focused+ UHPLC (Thermo Fisher Scientific, Germering, Germany) system equipped with an Accucore C18 column (150 × 2.1 mm; 2.6 µm). Eluent A was acetonitrile:water, (50:50, *v/v*) and eluent B was isopropanol:acetonitrile:water, (85:10:5, *v/v*), both containing 0.1% formic acid and ammonium formate (5 mmol/L). The column temperature was at 50 °C and the flow rate was of 300 µL/min. Lipids were separated using gradient elution: 0−20 min ramp from 10% to 86% B, 20−22 min ramp to 86% to 95% B, 22−26 min 95% B, and the column was then re-equilibrated at 10% B for 8 min.

LC was coupled on-line to a Q Exactive Plus Hybrid Quadrupole-Orbitrap Mass Spectrometer (Thermo Fisher Scientific, Bremen, Germany) operated in positive ion mode using data-dependent acquisition (DDA). The S-lens RF level was set to 35%. Capillary temperature was set to 300 °C, and the aux gas heater temperature was 370 °C. Sheath, aux, and sweep gas flow rates were set to 40, 10, and 1 arbitrary units, respectively. DDA settings: the full-scan mode (scan range *m/z* 400 to 1200) was acquired at a resolution of 140,000 (at *m/z* 200); automatic gain control target 3e6; maximum injection time, 100 ms. The MS/MS mode was acquired at a resolution of 17,500 top 15 most intense ions. Ions were fragmented using HCD fragmentation, stepped normalized collision energy (15, 20, 30), isolation width 1.2 Da, automatic gain control target 1 × 10^5^ maximum injection time 60 ms, and intensity threshold 3.3 × 10^3^. In both, case profile spectrum data were acquired.

Data were analyzed using Xcalibur (version 4.0) and quantification was performed with Lipostar (version 1.0.7) [37]. All graphs were created using GraphPad Prism (version 5.02) for Windows (GraphPad Software, San Diego California USA).

As absolute quantification of oxidized lipids by LC-MS would require the availability of a corresponding internal standard for each LPP and thus not possible so far, a relative quantification of oxidized species was performed in this study. To provided relative quantities for total lipid oxidation as well as for each individual LPP formed, the fold change between peak areas of oxidized lipid molecular species and peak area of the parent lipid in this particular preparation were calculated to account for the differences in initial concentrations of oxidizable lipids in liposomal preparations with different molar fractions of SM and PLPC.

## 4. Conclusions

The overall rate of lipid oxidation in biological membranes is determined by a complex set of factors including the composition of polyunsaturated fatty acyl chains of PLs, presence and ratio of radical trapping antioxidants (e.g., α-tocopherol and CoQ10), and membrane biophysical properties, such as fluidity and rigidity [13,14,15,16,17]. SM-rich membranes characterized by decreased fluidity and high rigidity due to the dense network of intra- and intermolecular H-bonds increase internal rigidity and intermolecular order [38,39]. This extended “hydrogen belt” was proposed to create a barrier for radical supply and propagation of lipid peroxidation [15]. SMs were proposed to play a role of membrane rigidifying lipids which block the contacts between adjacent PC lipids, and thus physically prevent lateral propagation of lipid radicals [25].

Here, we confirm the effect of liposomal SMs as inhibitors of lipid oxidation propagation. Furthermore, using electrochemical oxidation, we demonstrated that this is effect is eliminated in non-liposomal SM solutions, underlying the importance of membrane organization and H-bond network. We could further shed light on the mechanistic aspects of this inhibition by illustrating the shift in LPP patterns in liposomes with high SM fraction towards lower abundance of propagating mono-hydroxy (epoxy) species vs. corresponding keto-derivatives. When monitored by LC-MS, this keto/hydroxyl (epoxy) LPP ratio can be used in the future studies as a marker of propagation potential of free radical lipid peroxidation in various biological systems. Taken together with the previous studies on the inhibition of lipid peroxidation by SM in lipoproteins and model liposomes [24,25,26,27], the results presented here illustrate the important property of SMs in biological membranes, acting as “biophysical antioxidant”. SMs are enriched at the outer leaflet of plasma membrane where they account for up to 55 mol% relative to other phospholipids [40] and might provide an important protection to cellular membranes against oxidants produced by activated phagocytic cells, as well as a variety of pro-oxidative compounds. The role of SM in protecting polyunsaturated lipids of the inner membrane leaflet is not studied so far. Nevertheless, using different fibroblast culturing conditions, it was demonstrated that the higher content of SM in plasma membrane was protective against cell susceptibility to oxidative stress and this effect was eliminated by treatment with exogenous sphingomyelinase [27].

Considering the significance of SM-rich membrane microdomains (lipid rafts) in the regulation of membrane dynamics, activity of membrane proteins, and signal transduction, SM antioxidant properties might be crucial in local prevention of lipid peroxidation and associated protein modifications in these membrane substructures.

## Figures and Tables

**Figure 1 molecules-25-01925-f001:**
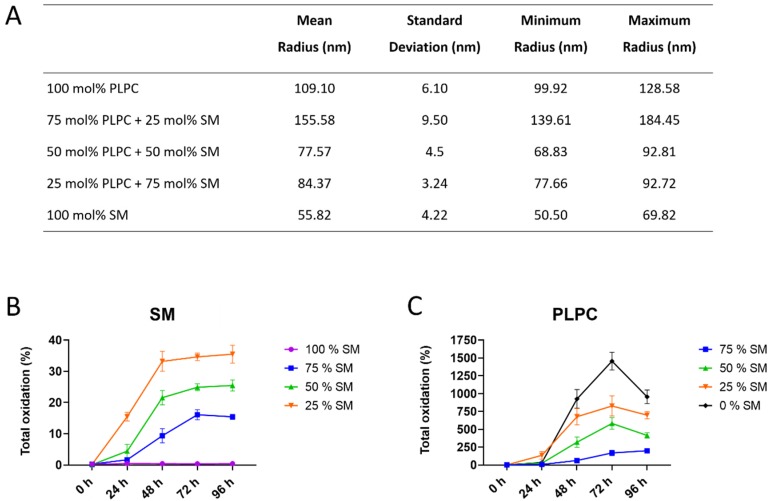
Characterization of model liposomes and the extent of total sphingomyelin (SM)- and glycerophosphatidylcholine (PC)-derived lipid peroxidation products produced over the time upon addition of Cu^2+^/ascorbate. (**A**) Radius of liposomes with different SM/PC molar ratio determined by dynamic light scattering; (**B**) SM and (**C**) 1-palmitoyl-2-linoleoyl-sn-glycero-3-phosphocholine (PLPC) total oxidation monitored for 96 h of incubation in the presence of Cu^2+^/ascorbate expressed as the sum of lipid peroxidation product (LPP) peak areas quantified in each sample relative to the peak area of unoxidized parent lipid.

**Figure 2 molecules-25-01925-f002:**
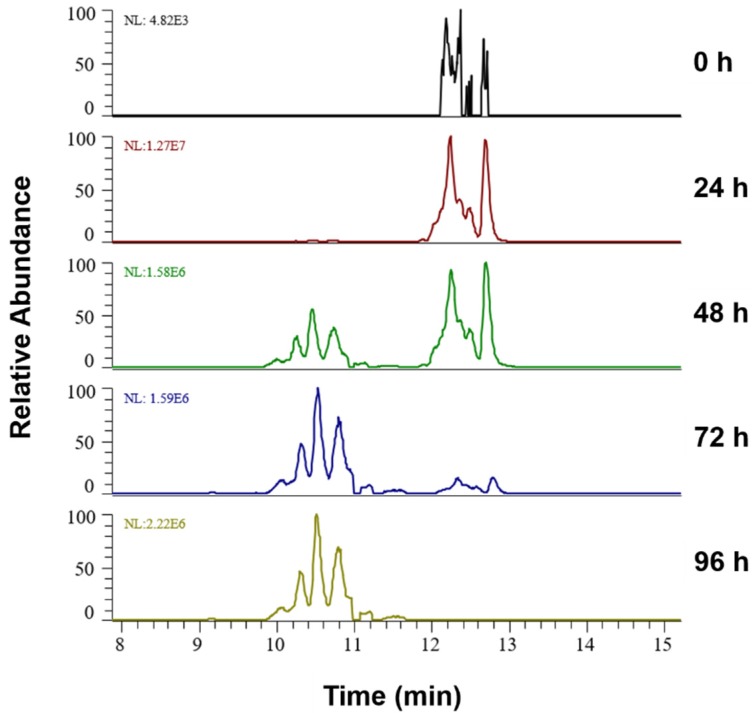
Extracted ion chromatograms for the signal at *m/z* 790.5598^+^ (±5 ppm; PLPC+2O) in liposomes containing 75 mol% SM over the oxidation time from 0 till 96 h, represented by two isomeric LPPs—dihydroxy (RT 10–11 min) and hydroperoxy (RT 12–13 min) PLPC.

**Figure 3 molecules-25-01925-f003:**
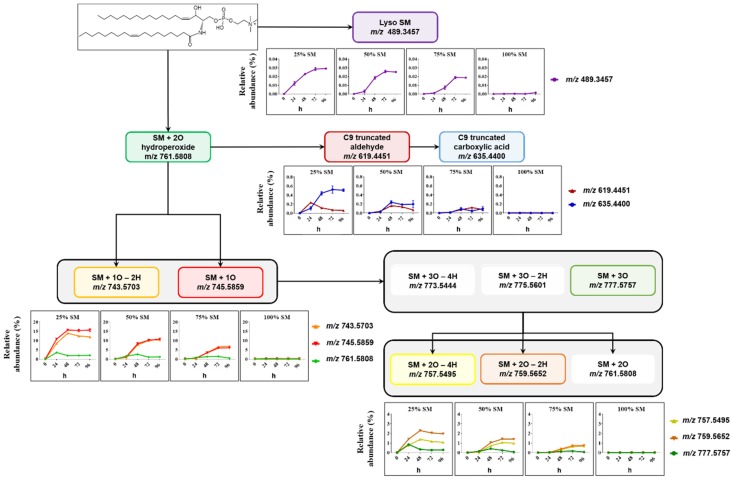
Overview of SM-derived LPPs quantified in liposomes with different SM/PC ratio oxidized in the presence of Cu^2+^/ascorbate for 96 h. Relative abundance of each LPP is calculated as the peak area of LPP divided by peak area of unmodified SM in the same sample.

**Figure 4 molecules-25-01925-f004:**
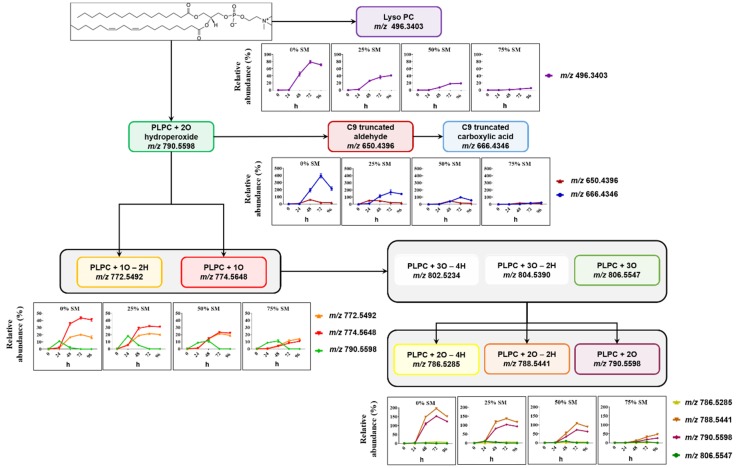
Overview of PLPC-derived LPPs quantified in liposomes with different SM/PC ratio oxidized in the presence of Cu^2+^/ascorbate for 96 h. Relative abundance of each LPP is calculated as the peak area of LPP divided by the peak area of unmodified PLPC in the same sample.

**Figure 5 molecules-25-01925-f005:**
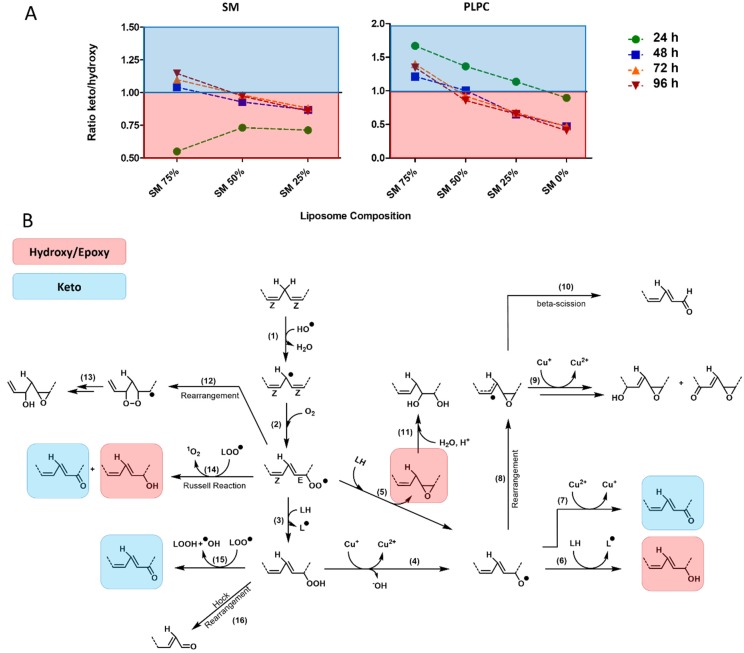
Ratio of keto to hydroxy(epoxy) mono-oxygenated SM and PLPC as a marker of lipid peroxidation propagation. (**A**) Calculated ratio for keto/hydroxy(epoxy) SM- and PLPC-derived LPPs in liposomes with different SM content illustrated for 24, 48, 72, and 96 h post oxidation induction. (**B**) Proposed pathways of free radical driven lipid peroxidation described for mono- and di-unsaturated fatty acids in the presence of transition metals. Mono-oxygenated keto- and hydroxy(epoxy)-derivatives are highlighted (blue and red, respectively).

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
