# Peer review of "Sphingomyelins Prevent Propagation of Lipid Peroxidation—LC-MS/MS Evaluation of Inhibition Mechanisms"

_molecules, 2020, doi:10.3390/molecules25081925_

Round 1

Reviewer 1 Report

In this manuscript, the authors investigated the mechanistic aspects of SM protective role against lipid peroxidation in liposomes containing a different ratio of SM and PLPC. The data generated during the study allowed the authors to demonstrate the redirection of the lipid peroxidation pathway from chain propagation, towards chain termination, depending on the SM levels. 

Globally, the manuscript is rather solid. The main comment I have is regarding the biological consequences of these results. SM and PC are at the outer leaflet, while the PUFA enriched phospholipids, as PE, PE-plasmalogens, PI, and PS are the inner membrane. Could the SM protect these fatty acids? In my opinion, this aspect should be taken into account during the discussion. 

Shorter questions that would need to consider are: 

  • Is there any evidence using a cell model that would support the results of this study? 
  • Figure 1 B. According to the authors when liposomes are constituted only by SM, they did not detect oxidated forms. However, in the introduction, they mentioned: “the possibility of formation of biologically relevant SM derived lipid peroxidation products (LPPs) cannot be excluded”. Taking into account that the SM(used has oleic acid (d18:1/18:1) as fatty acid, could the authors comment on why oxidized forms were not detected?

  • Figure 1C. Conceptually it would be more adequate to express the increase in oxidation as fold increase rather than in such high percentages.

Minor comments 

Page 2 Line 47 No need to abbreviate sphingoid base (SB), as it is not appearing anymore.

Line 75, PLPC is not defined until the Material and Methods section.

Author Response

In this manuscript, the authors investigated the mechanistic aspects of SM protective role against lipid peroxidation in liposomes containing a different ratio of SM and PLPC. The data generated during the study allowed the authors to demonstrate the redirection of the lipid peroxidation pathway from chain propagation, towards chain termination, depending on the SM levels.

Globally, the manuscript is rather solid. The main comment I have is regarding the biological consequences of these results. SM and PC are at the outer leaflet, while the PUFA enriched phospholipids, as PE, PE-plasmalogens, PI, and PS are the inner membrane. Could the SM protect these fatty acids? In my opinion, this aspect should be taken into account during the discussion.

To the best of our knowledge, nothing is known on the “antioxidant” effect of SM in plasma membrane to the inner leaflet unsaturated PLs. Nevertheless, it would be a very interesting point to be addressed in the future studies. Here we provided a short paragraph addressing Reviewer question in Conclusion of the revised manuscript:

“SM are enriched at the outer leaflet of plasma membrane where they account up to 55 mol% relative to other phospholipids [40] and might provide an important protection to cellular membranes against oxidants produces by activated phagocytic cells as well as variety of pro-oxidative compounds. Role of SM in protecting polyunsaturated lipids of inner membrane leaflet in not studied so far. Nevertheless, using different fibroblasts culturing conditions it was demonstrated that higher content of SM in plasma membrane was protective against cell susceptibility to oxidative stress and this effect was eliminated by treatment with exogenous sphingomyelinase [27].“

Shorter questions that would need to consider are:

Is there any evidence using a cell model that would support the results of this study?

See above

Figure 1 B. According to the authors when liposomes are constituted only by SM, they did not detect oxidated forms. However, in the introduction, they mentioned: “the possibility of formation of biologically relevant SM derived lipid peroxidation products (LPPs) cannot be excluded”. Taking into account that the SM(used has oleic acid (d18:1/18:1) as fatty acid, could the authors comment on why oxidized forms were not detected?

In our study we didn’t detect any oxidation in pure SM liposome which of course is an artificial system. In biological membranes, SM are present in a mixture with different other lipid species, thus their oxidation, however low, can be expected similar to the results obtained here with liposomes containing 25 and 50 mol% of SM lipids.

Figure 1C. Conceptually it would be more adequate to express the increase in oxidation as fold increase rather than in such high percentages

Following Reviewer suggestions, we tried to express our data in fold change values. However, after careful considerations we decided to stay with % increase, as fold change usually imply comparison of the exactly the same variable and here, we provide LPP increase or decrease relative to the associated change in peak area of parent lipid.

Minor comments

Page 2 Line 47 No need to abbreviate sphingoid base (SB), as it is not appearing anymore.

Corrected.

Line 75, PLPC is not defined until the Material and Methods section.

Corrected.

Reviewer 2 Report

The authors performed LC-MS/MS analysis of model SM/PC liposomes after radical driven lipid peroxidation to address the “antioxidative” potential (inhibition of peroxidation propagation) of SMs. Increasing amounts of SM in liposomes clearly suppressed lipid oxidation and confirmed the “antioxidative” potential of SMs. Since this effect was eliminated after electrochemical oxidation of non-liposomal SMs, the authors conclude, that membrane organization and H-bond networks are very important.

The manuscript is written in a clear form, the study is well-reported and important to the research field and of great interest to the readers of the journal.

However, the SM (d18:1/18:1) that was used in this study differs from PLPC not only by the ability to form hydrogen bonds, but also by significantly lower oxidation rate because of the presence of only single double bonds. In order to test, which of two differences is the basis for the “antioxidant” effect of SM, it is necessary to analyze kinetics of oxidation using PLPC liposomes containing increasing amounts of di-oleoyl-PC. In order to simplify the experiments, kinetics of oxidation in SM- and OOPC-containing liposomes can be compared using a spectrophotometric quantification of conjugated double bonds. The lack of antioxidant effect of OOPC will be an indication of the importance of H-bond formation.

Minor:

The method section of liposome preparation and characterization could be explained in more detail. e.g. total amount of lipids and volumes used during liposome preparation; molarity of lipids during liposome preparation; molarity of lipids for DLS analysis; sonication power (W/Hz) etc.

Line 168-170: please rephrase this sentence

Author Response

The authors performed LC-MS/MS analysis of model SM/PC liposomes after radical driven lipid peroxidation to address the “antioxidative” potential (inhibition of peroxidation propagation) of SMs. Increasing amounts of SM in liposomes clearly suppressed lipid oxidation and confirmed the “antioxidative” potential of SMs. Since this effect was eliminated after electrochemical oxidation of non-liposomal SMs, the authors conclude, that membrane organization and H-bond networks are very important.

The manuscript is written in a clear form, the study is well-reported and important to the research field and of great interest to the readers of the journal.

However, the SM (d18:1/18:1) that was used in this study differs from PLPC not only by the ability to form hydrogen bonds, but also by significantly lower oxidation rate because of the presence of only single double bonds.

In order to test, which of two differences is the basis for the “antioxidant” effect of SM, it is necessary to analyze kinetics of oxidation using PLPC liposomes containing increasing amounts of di-oleoyl-PC. In order to simplify the experiments, kinetics of oxidation in SM- and OOPC-containing liposomes can be compared using a spectrophotometric quantification of conjugated double bonds. The lack of antioxidant effect of OOPC will be an indication of the importance of H-bond formation.

Unfortunately, considering current situation with COVID-19 pandemic status and associated lockdowns of the university laboratories, it is not possible for us at the current state to perform experiment suggested. We resubmit our manuscript in the current form by providing additional discussion on the point questioned by Reviewer 1, which is in our view is sufficient to address this question.

“Some studies compared inhibitory effect of SM-rich liposomes on lipoxidation of unsaturated PC lipids with liposomes containing dipalmitoyl PC (DPPC). Interestingly, using absorbance of conjugated dienes as the read out of lipid oxidation, both SM and DPPC showed similar inhibition of PLPC oxidation [26]. However, when MS based method was used to monitor lipid oxidation of stearyl-arachidonoyl-PC (SAPC) liposomes, presence of SM resulted in much lower oxidation rates then DPPC, indicating importance of SM-specific H-bond network in preventing radical propagation [25].

 Here, using high-resolution MS allowing us to monitor the whole variety of LPPs independent on the presence of conjugated dienes in their structure, we could demonstrate an apparent shift in the pattern within formed LPPs.”

Minor: The method section of liposome preparation and characterization could be explained in more detail. e.g. total amount of lipids and volumes used during liposome preparation; molarity of lipids during liposome preparation; molarity of lipids for DLS analysis; sonication power (W/Hz) etc.

Additional details were added to the Methods section.

Line 168-170: please rephrase this sentence

Referred sentence was rephrased.

Reviewer 3 Report

This study aims at demonstrating a potential inhibitory effect of sphingomyelins (SM) on phospholipid (PLPC) acyl chain oxidation in liposomes, an effect that is expected to be representative of what might happen in real membrane systems. Lipid species are measured by mass spectrometry coupled to liquid chromatography. The experimental set-up and manuscript drawing are both excellent. However, some concerns on data management arise that warrant addressing or discussion by the authors. They are:

1) It can be expected that both oxidation rate (OR) and total content of oxidized species (TOS) are dependent upon the initial concentration of the species subjected to be oxidized. Data on figure 1 and Table S1 seem to show this in fact. Accordingly, which would actually be a clue on the potential inhibitory action of SMs is the variation of the constant in the equation [TOS] = α*[P]0, where α is the constant that measure the rate of oxidation (OR) and [P]0 is the inital concentration of a PLPC given species. How does α vary with [SM]? That is to say: is the concentration of SM which actually acts inhibiting or there is lower oxidation because of the lower concentration of PLPCs? If [PLPC] = 0 there may not be neither oxidation nor inhibition. Furthermore, if oxidized species of SM increase with the content of SM it does not mean inhibition but the most abundant species is the subject of the oxidation process, doesn’t it? Most mammalian tissues do not have more than 15-20% of SMs or even less.

2) The presentation of data as the ratio between [TOS] and [P], where [P] is the concentration of the parent PLPC species for TOS, is somewhat confusing and makes difficult the evaluation of results. It is being assumed that only one oxidized species is arising from a given PLPC species, but several species might be formed as there are two acyl chains that may undergo different ways of oxidation. According to the way used by the authors to express the results, it can be show [OS]/[P] = ([P]0 – [P])/[P] = ([P]0/[P]) – 1, where [OS] is the concentration of the oxidized species arising from P. Comparison would be much easier if data are presented as the percentage of the concentration of the not oxidized species in regards to the initial concentration of that given species ([P]/[P]0), or as the concentration of the oxidized species in regards to the initial related PLPC concentration. If there are more than one oxidized species arising from a given PLPC species (see lines 123-126 in the text) [OS] should be changed by [TOS] to avoid overfitting. 2OH oxidized species come from OH oxidized species and, hence, they should be considered as the same oxidized species; the same can be said for keto oxidized species as they have to undergo a first step of OH oxidation.

Author Response

This study aims at demonstrating a potential inhibitory effect of sphingomyelins (SM) on phospholipid (PLPC) acyl chain oxidation in liposomes, an effect that is expected to be representative of what might happen in real membrane systems. Lipid species are measured by mass spectrometry coupled to liquid chromatography. The experimental set-up and manuscript drawing are both excellent. However, some concerns on data management arise that warrant addressing or discussion by the authors. They are:

  • It can be expected that both oxidation rate (OR) and total content of oxidized species (TOS) are dependent upon the initial concentration of the species subjected to be oxidized. Data on figure 1 and Table S1 seem to show this in fact.

Accordingly, which would actually be a clue on the potential inhibitory action of SMs is the variation of the constant in the equation [TOS] = α*[P]0, where α is the constant that measure the rate of oxidation (OR) and [P]0 is the inital concentration of a PLPC given species. How does α vary with [SM]? That is to say: is the concentration of SM which actually acts inhibiting or there is lower oxidation because of the lower concentration of PLPCs? If [PLPC] = 0 there may not be neither oxidation nor inhibition. Furthermore, if oxidized species of SM increase with the content of SM it does not mean inhibition but the most abundant species is the subject of the oxidation process, doesn’t it? Most mammalian tissues do not have more than 15-20% of SMs or even less.

To address exactly this point in our original calculations, for relative quantification we used for each liposomal preparation the peak area of parent lipid as 100% to eliminate the effect of different original concentrations of parent lipids in liposomes with different mol fractions.

  • The presentation of data as the ratio between [TOS] and [P], where [P] is the concentration of the parent PLPC species for TOS, is somewhat confusing and makes difficult the evaluation of results. It is being assumed that only one oxidized species is arising from a given PLPC species, but several species might be formed as there are two acyl chains that may undergo different ways of oxidation.

According to the way used by the authors to express the results, it can be show [OS]/[P] = ([P]0 – [P])/[P] = ([P]0/[P]) – 1, where [OS] is the concentration of the oxidized species arising from P. Comparison would be much easier if data are presented as the percentage of the concentration of the not oxidized species in regards to the initial concentration of that given species ([P]/[P]0), or as the concentration of the oxidized species in regards to the initial related PLPC concentration. If there are more than one oxidized species arising from a given PLPC species (see lines 123-126 in the text) [OS] should be changed by [TOS] to avoid overfitting. 2OH oxidized species come from OH oxidized species and, hence, they should be considered as the same oxidized species; the same can be said for keto oxidized species as they have to undergo a first step of OH oxidation.

Due to the nature of mass spectrometry derived data, we unfortunately can not address all the changes in the studied system using absolute concentrations of parent and oxidized lipids. To comment on it, we provided additional explanations on quantification strategy used in the study in Method section:

“As absolute quantification of oxidized lipids by LC-MS would require availability of corresponding internal standard for each LPP and thus not possible so far, relative quantification of oxidized species was performed in this study. To provided relative quantities for total lipid oxidation as well as for each individual LPP formed, fold change between peak areas of oxidized lipid molecular species and peak area of the parent lipid in this particular preparation were calculated to account for the differences in initial concentrations of oxidizable lipids in liposomal preparations with different molar fractions of SM and PLPC.“

Round 2

Reviewer 2 Report

The extended discussion improved the quality of the manuscript.